# Diffusion-MRI-based regional cortical microstructure at birth for predicting neurodevelopmental outcomes of 2-year-olds

**Minhui Ouyang[1], Qinmu Peng[1,2], Tina Jeon[1], Roy Heyne[3], Lina Chalak[3], Hao Huang[1,2]***

[1]Radiology Research, Children's Hospital of Philadelphia, Philadelphia, United States; [2]Department of Radiology, Perelman School of Medicine, University of Pennsylvania, Philadelphia, United States; [3]Department of Pediatrics, University of Texas Southwestern Medical Center, Dallas, United States

**Abstract** Cerebral cortical architecture at birth encodes regionally differential dendritic arborization and synaptic formation. It underlies behavioral emergence of 2-year-olds. Brain changes in 0–2 years are most dynamic across the lifespan. Effective prediction of future behavior with brain microstructure at birth will reveal structural basis of behavioral emergence in typical development and identify biomarkers for early detection and tailored intervention in atypical development. Here we aimed to evaluate the neonate whole-brain cortical microstructure quantified by diffusion MRI for predicting future behavior. We found that individual cognitive and language functions assessed at the age of 2 years were robustly predicted by neonate cortical microstructure using support vector regression. Remarkably, cortical regions contributing heavily to the prediction models exhibited distinctive functional selectivity for cognition and language. These findings highlight regional cortical microstructure at birth as a potential sensitive biomarker in predicting future neurodevelopmental outcomes and identifying individual risks of brain disorders.

**\*For correspondence:**
huangh6@email.chop.edu

**Competing interests:** The authors declare that no competing interests exist.

## Introduction

Brain cerebral cortical microstructure underlies neuronal circuit formation and function emergence during brain maturation. Regionally distinctive cortical microstructural architecture profiles around birth result from immensely complicated and spatiotemporally heterogeneous underlying cellular and molecular processes (*Silbereis et al., 2016*), including neurogenesis, synapse formation, dendritic arborization, axonal growth, pruning, and myelination. Disturbance of such precisely regulated maturational events is associated with mental disorders (*Innocenti and Price, 2005*). Diffusion magnetic resonance imaging (dMRI) has been widely used for quantifying microstructural changes in white matter (WM) maturation (e.g. *Dubois et al., 2008*; *Mukherjee et al., 2001*). Because of its sensitivity to organized cortical tissue (e.g. radial glial scaffold; *Rakic, 1995*; *Sidman and Rakic, 1973*) unique in the fetal and infant brain, dMRI also offers insights into maturation of cortical cytoarchitecture. Cortical fractional anisotropy (FA), a dMRI-derived measurement, of infant and fetal brain can effectively quantify local cortical microstructural architecture related to dendritic arborization and synaptic formation. Thus, cortical FA can potentially be used to infer specific brain circuit formation. In early cortical development, most of cortical neurons are generated in the ventricular and subventricular zone. These neurons migrate toward the cortical surface along a radially arranged scaffolding of glial cells where relatively high FA values are usually observed (*Huang et al., 2013*; *McKinstry et al., 2002*). During emergence of brain circuits, increasing dendritic arborization

(*Bystron et al., 2008*; *Sidman and Rakic, 1973*), synaptic formation (*Huttenlocher and Dabholkar, 1997*), and myelination of intracortical axons (*Yakovlev and Lecours, 1967*) disrupt the highly organized radial glia in the immature cortex and result in cortical FA decreases. Such reproducible cortical FA change patterns were documented in many studies of perinatal human brain development (*Ball et al., 2013*; *Huang et al., 2006*; *Huang et al., 2009*; *Huang et al., 2013*; *Kroenke et al., 2007*; *McKinstry et al., 2002*; *Neil et al., 1998*; *Ouyang et al., 2019a*; *Ouyang et al., 2019b*; *Yu et al., 2016*), suggesting sensitivity of cortical FA measures to maturational processes of cortical microstructure. Diffusion-MRI-based regional cortical microstructure at birth, encoding rich 'footage' of regional cellular and molecular processes, may provide novel information regarding typical cortical development and biomarkers for neuropsychiatric disorders.

The first 2 years of life is a critical period for behavioral development, with brain development in this period most rapid across the lifespan. In parallel to rapid maturation of cortical architecture and establishment of complex neuronal connections (*Hüppi et al., 1998*; *Ouyang et al., 2019a*; *Pfefferbaum et al., 1994*), babies learn to walk, talk, and build the core capacities for lifetime. Infant behaviors including cognition, language, and motor emerge during this time and become measurable at around 2 years of age. Reliable diagnosis for many neuropsychiatric disorders, such as autism spectrum disorder (ASD), can be made only around 2 years of age or later (*Marín, 2016*), as diagnoses rely on observing behavioral problems that are difficult to recognize in early infancy (*Arpi and Ferrari, 2013*; *Ozonoff et al., 2010*). On the other hand, early intervention for ASD, especially before 2 years of age, has demonstrated potential on improving outcomes (*Rogers et al., 2014*). Given that infants cannot communicate with language or writing in early infancy, there may be no better way to assess their brain development other than neuroimaging. Prediction of future cognition and behavior at 2 years of age or later based on brain features around birth creates an invaluable time window for individualized biomarker detection and early tailored intervention leading to better outcomes.

Individual differences in brain WM microstructural architectures (*Scholz et al., 2009*; *Yu et al., 2020*), behavior, and functions (*Braga and Buckner, 2017*; *Xu et al., 2019*) have been well recognized. Individual variability in brain structures and associated individual variability in future behaviors can be harnessed for robust prediction at the single-subject level (*Kanai and Rees, 2011*; *Rosenberg et al., 2018*), a step further than group classification. A few studies have been conducted previously to investigate within-sample imaging-outcome correlations (*Ball et al., 2015*; *Counsell et al., 2014*; *Deoni et al., 2016*; *Hintz et al., 2015*; *Keunen et al., 2017*; *Peyton et al., 2020*; *Wee et al., 2017*; *Woodward et al., 2006*), while such correlation approaches made it impossible to be applied to new and incoming subjects. Machine learning approaches that can adopt new subjects and yield continuous prediction values have been explored only recently based on WM structural networks (*Girault et al., 2019*; *Kawahara et al., 2017*). Compared to association of WM microstructure with cortical regions through WM fiber end point connectivity, association of cortical microstructure with cortical regions is more direct. Thus, FA of a specific cortical region can be used to directly reflect certain functions of the same cortical region. Our previous study *Ouyang et al., 2019b* demonstrated that cortical FA predicted neonate age with high accuracy. Regionally distinctive cortical microstructure around birth encodes the information that may predict distinctive functions manifested by future behavior and potentially identify the most sensitive regions as imaging markers to detect early behavioral abnormality. However, dMRI-based cortical microstructure has not been evaluated for predicting either discrete or continuous future behavioral measurement so far. And dMRI-based cortical microstructure has not been incorporated into a machine-learning-based prediction model for predicting future behavior, either.

In this study, we leveraged individual variability of cortical microstructure profiles of neonate brains for predicting future behavior. A novel machine-learning-based model using regional cortical microstructure markers from dMRI was developed to predict continuous outcome values. This model is also capable of incorporating new subjects in contrast to within-sample imaging-outcome correlations. We hypothesized that dMRI-based cortical microstructure at birth only (without inclusion of any WM microstructure information) could robustly predict the future neurodevelopmental outcomes. Out of 107 recruited neonates, high-resolution ($0.656 \times 0.656 \times 1.6$ mm$^3$) dMRI data were acquired from 87 neonates, of which 46 underwent a follow-up study at their 2 years of age for neurobehavioral assessments of cognitive, language, and motor abilities. Cortical microstructural architectures at birth were quantified by cortical FA on the cortical skeleton to alleviate partial volume

effects (*Ouyang et al., 2019b*; *Yu et al., 2016*). Regional cortical FA measures were then used to form feature vectors to predict neurodevelopmental outcomes at 2 years of age. We further quantified the contribution of each cortical region in predicting different outcomes, as distinctive behaviors are likely encoded in uniquely distributed pattern across the cerebral cortex.

## Results

### Cortical microstructure at birth and neurodevelopmental outcomes at 2 years of age

A cohort of 107 neonates was recruited for studying prenatal and perinatal human brain development (see more details in Materials and methods and *Supplementary file 1*). Neuroimaging data, including structural and diffusion MRI, were collected from 87 infants around birth in their natural sleep. Forty-six infants went through a follow-up visit at their 2 years of age to complete the cognitive, language, and motor assessments with Bayley scales of infant and toddler development-Third Edition (Bayley-III; *Bayley, 2006*). *Figure 1—figure supplement 1* and *Figure 1—figure supplement 2* demonstrate the cortical FA maps across parcellated cortical gyri in the left and right hemisphere from dMRI of all these 46 subjects scanned at birth, revealing individual variability of regional cortical microstructure. The Bayley-III composite scores from these 46 subjects at 2 years of age range from 65 to 110 (mean ± sd: 87.4 ± 8.5) for cognition, 56 to 112 (85.7 ± 10.1) for language, and 73 to 107 (91.2 ± 7.1) for motor abilities (*Figure 1—figure supplement 3a*). No significant differences between preterm and term born infants were found in any of the Bayley-III composite scores (all p>0.3; *Figure 1—figure supplement 3b*). No significant correlation between any specific age (i.e. birth age, MRI scan age, and Bayley-III exam age) and neurodevelopmental outcome score was found in either preterm or term born infant groups (all p>0.1; *Supplementary file 2*).

### Robust prediction of cognitive and language outcomes based on cortical dMRI measurement

Fifty-two cortical regions parcellated by transforming neonate atlas labels (*Figure 1—figure supplement 4*) were used to generate cortical FA feature vectors from each participant's dMRI data at birth, representing the entire cortical microstructural architecture of an individual neonate (*Figure 1*; see Materials and methods). Heterogeneous cortical FA values distributed across cortical regions can be appreciated from cortical FA maps (left panels in *Figure 1*), indicating regionally differentiated maturation level of cortical microstructure. An immature cerebral cortical region with highly organized radial glia scaffold is associated with high FA values, whereas a more mature cortical region with extensive dendritic arborizations and synapses formations is associated with low FA values. To determine whether cortical microstructural features represented by FA measurements at birth are capable of predicting neurodevelopmental outcomes of an individual infant at a later age, we used support vector regression (SVR) with a fully leave-one-out cross-validated (LOOCV) approach (middle panels of *Figure 1*). With this approach, the neurodevelopmental outcome of each infant was predicted from an independent training sample. That is, for each testing subject out of the 46 participants, the cortical FA features of remaining 45 subjects were used to train prediction models for predicting cognitive or language outcomes of the testing subject at 2 years of age (right panels in *Figure 1*) only based on cortical FA of the testing subject at birth. An SVR model that best fits the training sample can be represented by a weighted contribution of all features, where the weight vector ($\vec{w}$) indicates the relative contribution of each feature, namely, cortical FA of each parcellated cortical region, to the prediction model. The feature contribution weights in the model predicting cognition or language were averaged across all leave-one-out SVR models and then normalized to $|w_i|/\sum|w_i|$ with $i$ indicating $i^{th}$ cortical gyrus. These normalized feature contribution weights were projected back onto the cortical surface to demonstrate cortical regional contribution (right panels in *Figure 1*).

Significant correlations between predicted and actual neurodevelopmental outcome were found for both cognitive ($r = 0.536$, p=$1.2 \times 10^{-4}$) and language ($r = 0.474$, p=$8.8 \times 10^{-4}$) scores, respectively (left panel in *Figure 2a and b*), indicating robust prediction of cognitive and language outcomes at 2 years old based on cortical FA measures at birth. According to the permutation tests, these correlations were significantly higher than those obtained by chance (p<0.005). The mean

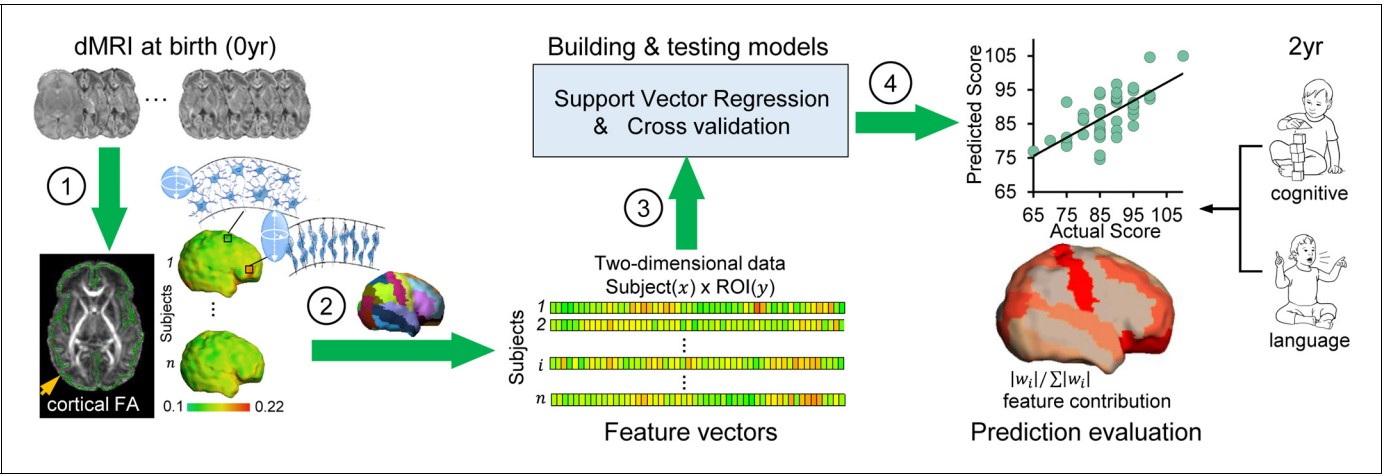

**Figure 1.** Workflow of predicting neurodevelopmental outcomes at 2 years based on cortical microstructural architecture at birth. Cortical microstructure at birth (0 year) quantified with cortical fractional anisotropy (FA) measures from diffusion magnetic resonance imaging (dMRI) was used to predict cognitive and language abilities assessed with Bayley-III Scales at 2 years of age (2 year). The prediction workflow includes the following steps: (1) Cortical microstructure was measured at the 'core' of cortical mantle, shown as green skeleton overlaid on a FA map and projected on a neonate cortical surface, to alleviate the partial volume effects. Schematic depiction of dendritic arborization and synaptic formation underlying cortical FA decreases during cortical microstructural maturation is shown. (2) Feature vectors were obtained by measuring cortical skeleton FA at parcellated cortical gyri with the gyral labeling transformed from a neonate atlas. Each parcellated cortical gyrus is a region-of-interest (ROI). (3) Prediction models were established and tested with support vector regression (SVR) and cross-validation. Feature vectors from all subjects were concatenated to obtain the input data of prediction models. (4) Prediction model accuracy was evaluated by correlation between predicted and actual scores. Feature contributions from different gyri in the model were quantified by normalized feature contribution weights which were projected back on a cortical surface for visualization.

The online version of this article includes the following figure supplement(s) for figure 1:

**Figure supplement 1.** Cortical fractional anisotropy (FA) distribution across parcellated cortical gyri in the left hemisphere from 46 infants who also went through neurodevelopmental assessments with Bayley-III at their 2 years of age.

**Figure supplement 2.** Cortical fractional anisotropy (FA) distribution across parcellated cortical gyri in the right hemisphere from 46 infants who also went through neurodevelopmental assessments with Bayley-III at their 2 years of age.

**Figure supplement 3.** Bayley-III scores of the studied population.

**Figure supplement 4.** The workflow to parcellate the cortical skeleton and measure the fractional anisotropy (FA) values at the parcellated cortical skeleton from a representative preterm (33 postmenstrual weeks, PMW).

**Figure supplement 5.** Overall small motion in diffuson magnetic resonance imaging scans of the studied population.

absolute errors (MAEs) between the predicted and actual scores are 5.49 and 7 for cognitive and language outcomes respectively. These MAEs were significantly lower than those obtained by chance (p<0.01), based on permutation tests. The highly predictive models suggest that cortical microstructural architecture at birth plays an important role in predicting future behavioral and cognitive abilities. However, motor scores were not able to be predicted from cortical FA measures ($r = 0.1$, p=0.52).

## Evaluation of robustness of prediction models

### Evaluation with different cortical parcellation schemes and age effects around birth

The prediction models are robust based on evaluation results of different cortical parcellation schemes; and the prediction results are still significant after age adjustment in the cortical FA features (*Figure 2—figure supplement 1*). To investigate the effects of different cortical parcellation schemes on prediction models, we measured regional cortical FA values with different cortical parcellation schemes that included higher number (128, 256, 512, and 1024) of random cortical parcels. For each parcellation scheme, we calculated correlation coefficient and MAE between the actual and predicted neurodevelopmental scores shown in *Figure 2—figure supplement 1*. Across different cortical parcellation schemes, robust estimation of the cognitive and language scores was observed in all prediction models. We also investigated the effect of different scan ages on prediction models

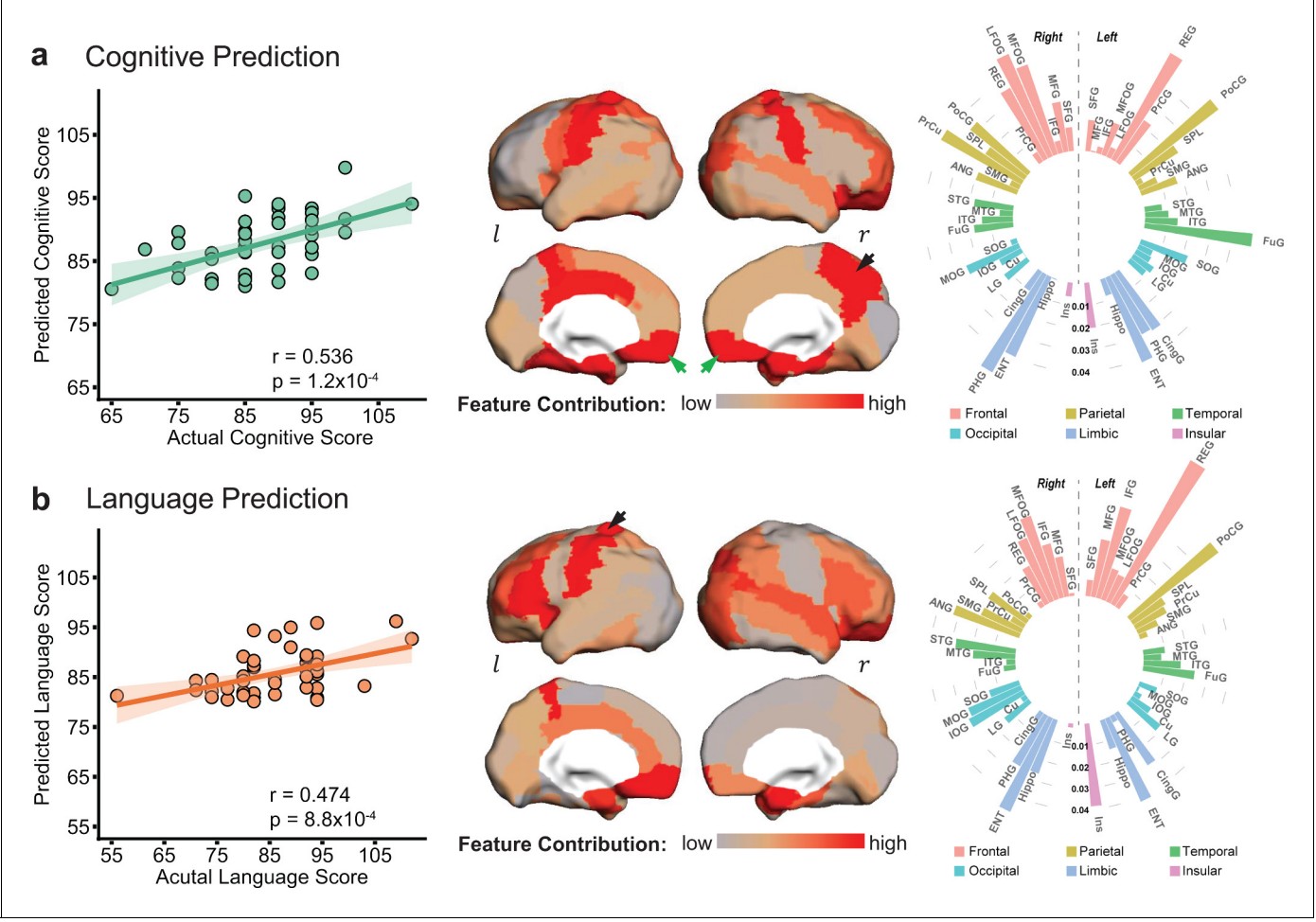

**Figure 2.** Cortical microstructural measures from neonate diffusion magnetic resonance imaging predict cognitive (a) and language (b) scores at 2 years of age with different feature contribution weights from various cortical gyri. *Left panels:* The scatter plots show significant correlation between actual scores and cognitive ($r = 0.536$, p=$1.2 \times 10^{-4}$) or language ($r = 0.474$, p=$8.8 \times 10^{-4}$) scores predicted based on cortical fractional anisotropy measures. Each dot represents one subject and linear regression was used to assess predictive accuracy of the model. The width of the line denotes the 95% confidence interval around the linear model fit between predicted and observed scores. *Center panels:* Normalized feature contribution weights of all cortical gyri in the prediction models are projected on a cortical surface. *Right panels:* Normalized feature contribution weights from all cortical gyri are demonstrated in the circular bar. These gyri were grouped into frontal, parietal, temporal, occipital, limbic, and insular cortex. Abbreviations: *r*: right hemisphere; *l*: left hemisphere. See *Supplementary file 3* for abbreviations of cortical regions and values of normalized feature contribution weights from all cortical gyri.

The online version of this article includes the following figure supplement(s) for figure 2:

**Figure supplement 1.** Evaluation of robustness of prediction models with different cortical region-of-interest (ROI) schemes and age adjustments.

**Figure supplement 2.** Evaluation of the prediction models.

to demonstrate that high prediction performance remained intact after statistically controlling for the age effect in cortical FA measures. Prediction performances before and after adjustment for the age effect are demonstrated in *Figure 2—figure supplement 1b and 1c*. After adjustment for the age effect, correlation between the predicted and actual cognitive or language scores is still significant (p<0.05) with original parcellation of 52 cortical regions. Furthermore, significant correlations after controlling for age effect were also observed across other tested cortical parcellation schemes (128, 256, 512, and 1024 cortical parcels).

## Evaluation by categorizing subjects with normal and low scores

As Bayley-III is widely used to assess developmental delay with certain cut-off scores, we also evaluated the performance of cortical microstructural measures in classifying subjects with normal and

low scores. High accuracy was achieved with a receiver operating characteristic (ROC) curve analysis. Cognitive and language scores of all infants were categorized into normal (>85, n = 22 for cognitive scores and n = 24 for language scores) and low (≤85) score groups. Cortical FA features were used to build classifiers with leave-one-out procedure to classify each infant into one of these two groups. The ROC curve analysis was used to test the ability of cortical FA measures at birth to distinguish infants with low 2-year-old outcomes from those with normal outcomes (*Figure 2—figure supplement 2a*). Classification accuracy was 76.1% for cognitive and 60.9% for language scores (*Figure 2—figure supplement 2b*). Our analysis revealed an area under curve (AUC) of 0.809 and 0.737 for cognitive and language classifications (*Figure 2—figure supplement 2c*), respectively, supporting the cortical microstructural architecture at birth as a sensitive marker for prediction and potential detection of early behavioral abnormality at a population level.

## Regionally heterogeneous contribution to the cognitive and language prediction

Regional cortical FA measures across entire cortex did not contribute equally to the prediction models. Heterogeneous feature contribution pattern can be clearly seen across cortex for either cognitive (*Figure 2a*) or language (*Figure 2b*) prediction. For instance, from the distribution of normalized feature contribution weights in the cognitive prediction model (center panel in *Figure 2a*), high contributions from right precuneus gyrus (PrCu) (indicated by black arrow) and bilateral rectus gyri (REC) (indicated by green arrows) are prominent, with bright red gyri associated with high feature contribution. To quantitatively demonstrate heterogeneous feature contribution of all cortical gyri, the normalized feature contribution weights from 52 cortical gyri categorized into six cortices are shown in a circular bar plot (right panel in *Figure 2a*). Higher bar indicates higher feature contribution of a cortical region to the model. The normalized feature contribution weights of the frontal, parietal, and limbic gyri (e.g. REC, postcentral, and entorhinal gyri) are relatively higher than those of the occipital, temporal, and insular cortex (e.g. superior temporal or occipital gyri) in cognitive prediction.

Similar to cognitive prediction, regional variations of feature contribution can be observed in language prediction model, as demonstrated in cortical surface map and circular bar plot in *Figure 2b*. For example, higher feature contribution weight was found in the left postcentral gyrus (PoCG) (indicated by black arrow) than its counterpart in the right hemisphere. Feature contribution weights in the frontal and limbic gyri are also higher than those in the occipital and temporal gyri. Differential normalized feature contribution weights in cognitive or language prediction model across all cortical gyri are listed in *Supplementary file 3*.

## Distinguishable regional contribution to predicting cognitive or language outcomes

Besides regionally heterogeneous contributions, distinguishable feature contribution patterns were found in predicting cognitive or language outcomes. The top 10 cortical regions where microstructural measures contributed most to the prediction of cognitive and language scores are listed in *Figure 3a* and mapped onto cortical surface in *Figure 3b*. Among these top 10 cortical regions, left REC, bilateral entorhinal gyrus (ENT), right middle/lateral fronto-orbital gyrus (MFOG/LFOG), and left PoCG are the common regions (highlighted in yellow in *Figure 3b*) for predicting both cognitive and language outcomes. Right REC, right PrCu, right parahippocampal gyrus (PHG), and left fusiform gyrus (FuG) are unique to cognitive prediction (highlighted in red in *Figure 3b*), and left inferior frontal gyrus (IFG), left cingular gyrus (CingG), left insular cortex (INS), and right angular gyrus (ANG) are unique to language prediction (highlighted in green in *Figure 3b*). It is striking that left IFG, usually known as 'Broca's area' for language production, was uniquely found in the top contributing regions in language prediction model. Notably right PrCu, an important hub for default mode network, was uniquely found among the top contributing regions in cognitive prediction model. Bootstrapping analysis indicated that the top 10 cortical regions (*Figure 3*) where microstructural measures contributing most to prediction were highly reproducible from 1000 bootstrap resamples for predicting each behavioral outcome (*Figure 3—figure supplement 1*). As shown in *Figure 3—figure supplement 1*, the cortical regions with higher percentages (indicating higher reproducibility) in red or brown color overlap with the top 10 cortical regions where microstructural measures

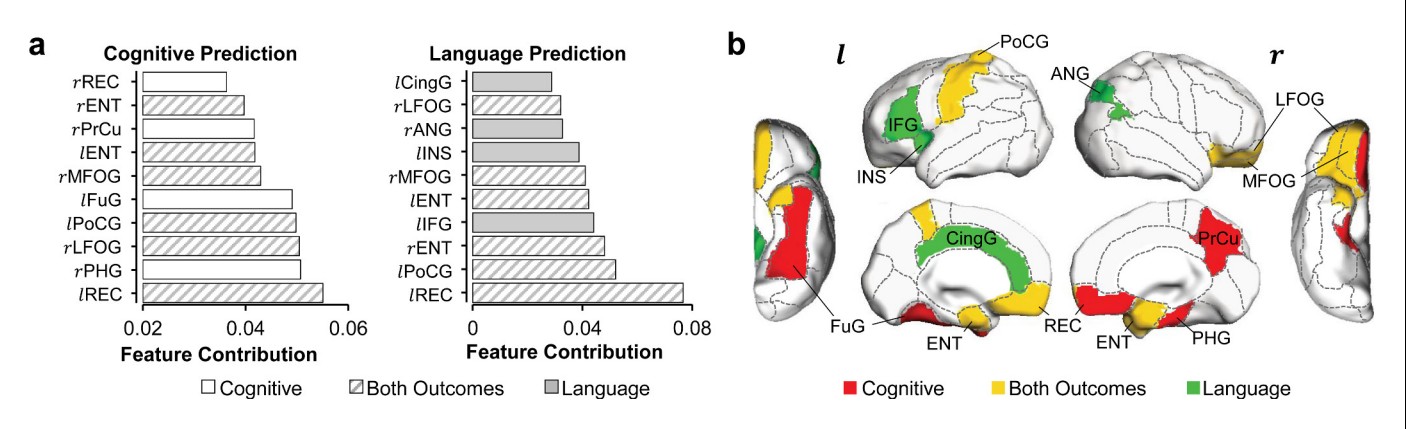

**Figure 3.** Distinguishable top 10 cortical regions where microstructural measures contributed most to the prediction of cognitive or language scores. (a) List of top 10 cortical regions with highest feature contribution weights in predicting cognitive (left) or language (right) scores. (b) Maps of cortical regions listed in (a). Cortical regions contributing most to predicting both cognition and language are painted in yellow (rMFOG, rLFOG, lREC, lPoCG, rENT, and lENT); Cortical regions contributing most to predicting uniquely cognition and language are painted in red (rREC, rPrCu, lFuG, and rPHG) and in green (lIFG, lCingG, rANG, and lINS), respectively. Abbreviations: *r*: right hemisphere; *l*: left hemisphere; ANG: angular gyrus; CingG: cingular gyrus; ENT: entorhinal gyrus; FuG: fusiform gyrus; IFG: inferior frontal gyrus; INS: insular cortex; LFOG: lateral fronto-orbital gyrus; MFOG: middle fronto-orbital gyrus; PHG: parahippocampal gyrus; PoCG: postcentral gyrus; PrCu: precuneus gyrus; REC: rectus gyrus.

The online version of this article includes the following figure supplement(s) for figure 3:

**Figure supplement 1.** Percentage maps for evaluating reproducibility of top 10 cortical regions with highest feature contribution to predicting cognition (a) and language (b) outcomes from bootstrap analysis.

contributed most to predicting cognition or language (from *Figure 3*; highlighted by dashed blue contours). Distinguishable regional contribution to predicting different outcomes (cognition or language) was quantified by a nonoverlapping index, ranging from 0 to 1 with one indicating completely distinctive regions and 0 indicating same regions. The statistical significance of the observed nonoverlapping index 0.57 was confirmed with permutation tests. Specifically, the permutation tests indicated that the observed nonoverlapping index of 0.57 was not likely to be obtained by chance from predicting the same outcome (p=0.001 from testing with leave-one-out resamples; p=0.05 from testing with resamples by randomly selecting 90% of samples), supporting distinguishable regional contribution to predicting cognitive or language outcomes.

## Comparison among prediction based on cortical FA, WM FA, and combined cortical and WM FA

Since dMRI has been conventionally used mainly for measuring WM microstructure, we also evaluated the prediction performance using regional WM FA measures only as features (left panel in *Figure 4a*). Both cognitive ($r = 0.516$, p=$2.4 \times 10^{-4}$) and language ($r = 0.517$, p=$2.3 \times 10^{-4}$) scores can be reliably predicted with WM FA measures, indicated by significant correlations between predicted and actual scores (right panel in *Figure 4a*). More importantly, solely cortical FA measures at birth are as robust as WM FA measures in predicting the cognitive and language scores at 2 years of age, demonstrated by similar correlation coefficient values between the predicted and actual scores (*Figure 4c*, *all* significant correlation with p<0.05). Combined cortical and WM FA measures as features (left panel in *Figure 4b*) showed higher performance with LOOCV than solely cortical or WM FA measures (*Figure 4c*) in predicting cognitive ($r = 0.721$, p=$1.6 \times 10^{-8}$) and language ($r = 0.614$, p=$5.6 \times 10^{-6}$) scores (right panel in *Figure 4b*). Combined features for prediction also passed threefold cross validation with significant correlation between predicted and actual cognitive ($r = 0.635$, p=0.01) and language ($r = 0.592$, p=0.02) scores across folds.

We further examined the correlation between motion (quantified by mean framewise displacement) and regional FA values from cerebral cortex and WM (*Figure 4—figure supplement 1a*). After false discovery rate (FDR) correction, no significant correlation between motion estimates and FA values of any of 92 brain regions was found. We also investigated motion effects on prediction results and found that high performance of prediction models was not affected after statistically

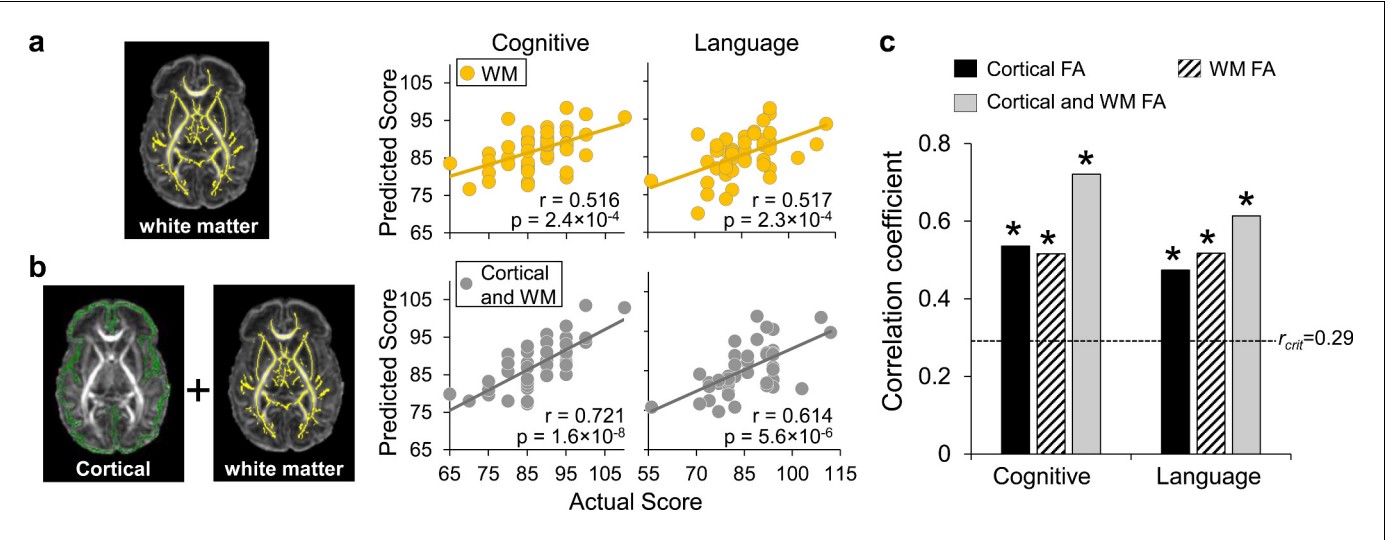

**Figure 4.** Prediction of neurodevelopmental outcomes using (**a**) white matter (WM) fractional anisotropy (FA) features only and (**b**) combined cortical and WM FA features, compared with prediction using cortical FA features. (**a**) On the left panel, WM microstructure was measured at the core WM regions, shown as yellow skeleton overlaid on a FA map of a neonate brain, to alleviate the partial volume effects. Feature vectors were obtained by measuring WM skeleton FA at 40 tracts transformed from WM labeling of a neonate atlas. On the right panel, scatter plots show linear regressions between actual scores and the predicted cognitive or language scores based on WM FA measures with LOOCV. (**b**) On the left panel, feature vectors were obtained by combining cortical skeleton (green) FA at 52 parcellated cortical gyri and WM skeleton (yellow) FA at 40 tracts with the labeling transformed from a neonate atlas. On the right panel, scatter plots show linear regressions between actual scores and the predicted cognitive or language scores based on both cortical and WM FA measures with LOOCV. (**c**) Significant correlation between the predicted and actual cognitive or language outcomes was found based on cortical FA only, WM FA only, and combined cortical and WM FA feature vectors. Dashed line indicates critical *r* value corresponding to p=0.05. * in the panel indicates significant (p<0.05) correlation.

The online version of this article includes the following figure supplement(s) for figure 4:

**Figure supplement 1.** Comprehensive motion-related analyses collectively demonstrate that motion did not confound the significant prediction found in this study.

controlling for motion estimates in cortical and WM FA measures in the predication models, as shown in *Figure 4—figure supplement 1b*. *Figure 4—figure supplement 1a and b* collectively demonstrated that motion did not contaminate or confound the significant predication found in this study.

## Discussion

We leveraged individual variability in the cortical microstructural architecture at birth for a robust prediction of future behavioral outcomes in continuous values. Cortical microstructure at birth, encoding rich 'footage' of regional cellular and molecular processes in early human brain development, was evaluated as the baseline measurements for prediction. Our previous studies (*Huang et al., 2009*; *Ouyang et al., 2019b*; *Yu et al., 2016*) found that individual neonate cortical microstructure profile characterized by different levels of dendritic arborization could be reliably quantified with dMRI-based cortical FA that reveals cortical maturation signature. In this study, we further demonstrated that individual variability of cortical microstructure profile around birth can be used to robustly predict the cognitive and language outcomes of individual infant at 2 years of age. By harnessing different encoding patterns of cognitive and language functions across the entire cortex, we quantified distinguishable contributions of each cortical region and highlighted the most sensitive regions for predicting different outcomes. Cortical regions contributing heavily to the prediction models exhibited distinctive functional selectivity for cognition and language. To our knowledge, this is the first study evaluating regional cortical microstructure for predicting future behavior, laying the foundation for future works using cortical microstructure profile as 'neuromarkers' to predict the risk of an individual developing health-related behavioral abnormalities (e.g. ASD). The

prediction model is also capable of incorporating new and incoming subjects, a step further than previous within-sample imaging-outcome correlation studies (*Ball et al., 2015*; *Counsell et al., 2014*; *Deoni et al., 2016*; *Hintz et al., 2015*; *Keunen et al., 2017*; *Peyton et al., 2020*; *Wee et al., 2017*; *Woodward et al., 2006*). Before the presented prediction can be used to identify individual infant at risk of neurodevelopmental disorders (e.g. ASD) at a time when the infant is pre-symptomatic in behavioral assessments, the prediction model needs to be further refined with a larger sample size.

Cerebral cortex plays a central role in human cognition and behaviors. High performance achieved in prediction of cognition and language based on cortical FA measures (*Figure 2*) is probably due to sensitivity of cortical microstructural changes to maturational processes including synaptic formation, dendritic arborization, and axonal growth. Distinctive maturation processes manifested by differentiated cortical FA changes across cortical regions in fetal and infant brains were reproducibly reported development (*Ball et al., 2013*; *Huang et al., 2006*; *Huang et al., 2009*; *Huang et al., 2013*; *Kroenke et al., 2007*; *McKinstry et al., 2002*; *Neil et al., 1998*; *Ouyang et al., 2019a*; *Ouyang et al., 2019b*; *Yu et al., 2016*). Although cortical thickness, volume, or surface area from structural MRI scans (i.e. T1- or T2-weighted images) were conventionally primary structural measurements to characterize infant cerebral cortex development (*Hazlett et al., 2017*; *Hill et al., 2010*; *Lyall et al., 2015*), they cannot characterize the complex microstructural processes that take place inside the cortical mantle. Compared to macrostructural changes quantified by these conventional measurements, the underlying microstructural processes quantified by cortical FA may be more sensitive to infants with pathology such as those with risk of ASD. Because WM microstructure (e.g. FA) measurement is more widely used in dMRI studies than cortical microstructure measurement, we also evaluated WM FA at birth for predicting cognitive and language outcome at 2 years of age (*Figure 4*). Despite the fact that microstructural measures from solely cerebral cortex or WM at birth have similar sensitivity in predicting neurodevelopment outcomes at 2 years old (*Figure 4*), cortical microstructure is more directly associated with specific cortical regions and thus certain cortical functions, compared to association of WM with the cortical regions through end point connectivity. On the other hand, combined cortical and WM microstructural measures at birth offered more baseline information and resulted in improved prediction (*Figure 4b and c*).

Human brain development in the first 2 years is most rapid across the lifespan. During first 2 years after birth, overall size of an infant brain increases dramatically, reaching close to 90% of an adult brain volume by 2 years of age (*Pfefferbaum et al., 1994*). Despite rapid development, infancy period (0–2 years) of human is probably the longest among all mammals with cognitive and language functions unique in human emerging during this critical period. For instance, infants start to learn their mother tongue from babbling to full sentences, during the age of 6 months to 2–3 years (*Kuhl, 2004*). This lengthy yet extremely dynamic brain development processes make the prediction of 2-year neurodevelopmental outcome with brain information at birth ultimately invaluable. DMRI-based cortical microstructure at birth, before any behavioral tests could be performed, can well predict both cognition and language outcomes at 2 years of age (*Figure 2*). A regionally heterogeneous distribution pattern across cerebral cortex was displayed (*Figure 2*). Furthermore, consistent with their functions documented in the literature, the cortical regions contributing heavily to the prediction models exhibited distinguishable functional selectivity for cognition and language (*Figure 3* and *Figure 3—figure supplement 1*). The cognitive scale in Bayley-III estimates cognitive functions including object relatedness, memory, problem solving, and manipulation on the basis of nonverbal activities (*Bayley, 2006*). Cortical regions with high weight in the prediction model are tightly associated with cognitive functions. Right rectus, precuneus, parahippocampal, and left fusiform gyri were among the top 10 cortical regions (painted red in *Figure 3b*) where microstructural measures contributed uniquely to the cognition prediction, but not to the language prediction. Parahippocampal gyrus provides poly-sensory input to the hippocampus (*Witter et al., 2000*) and holds an essential position for mediating memory function (*Young et al., 1997*). Precuneus is a pivotal hub essential in brain's default-mode network (*Buckner et al., 2008*) and involved in various higher-order cognitive functions (*Cavanna and Trimble, 2006*). Rectus and fusiform gyri are associated with higher-level social cognition processes (*Viskontas et al., 2007*). For regions contributing heavily to both cognition and language prediction and painted yellow in *Figure 3b*, bilateral entorhinal gyri likely serve as a pivot junctional region mediating the processes of different types of sensory information during the cortex-hippocampus interplay (*Witter et al., 2000*). Middle and lateral fronto-orbital gyrus are

involved in cognitive processes including learning, memory, and decision-making (*Wikenheiser and Schoenbaum, 2016*). Left postcentral gyrus with high feature contribution weight in both prediction models is associated with the general motor demands of performing tasks. Besides higher-order cognitive functions, language functions are also unique in human beings. The first 2 years of life is a critical and sensitive period for the speech-perception and speech-production development (*Kuhl, 2004*; *Werker and Hensch, 2015*). The language scale from Bayley-III includes two subdomains: receptive communication and expressive communication. Here the term 'communication' refers to any way that a child uses to interact with others, and includes communication in prelinguistic stage (e.g. eye gaze, gesture, facial expression, vocalizations, and words), social-emotional skills, and communication in more advanced stage of language emergence (*Bayley, 2006*). The distinctive regions with high feature contribution weights only in language prediction model included left inferior frontal gyrus (IFG), cingular gyrus, insular cortex, and right angular gyrus, painted green in *Figure 3b*. These regions are related to receptive and expressive communication. It is striking that left IFG, known as 'Broca's area', was identified by this data-driven prediction model because it is well known that Broca's area plays a pivotal role in producing language (*Poeppel, 2014*). Angular gyrus, another crucial language region in the parietal lobe, supports the integration of semantic information into context and transfers visually perceived words to Wernicke's area. Left insula, a part of the articulatory network in the dual-stream model of speech processing, is involved in translating acoustic speech signals into articulatory representations in the frontal lobe (*Hickok and Poeppel, 2007*). Since the Bayley language scale also includes a number of items reflecting social-emotional skills, such as how a child responds to his/her name or reacts when interrupted in play, the high feature contribution weight of insular cortex may be due to its important role in social emotions (*Lamm and Singer, 2010*). The high feature contribution weight of cingulate cortex might be related to its key involvement in emotion and social behavior (*Bush et al., 2000*). Taken together, identifying these regions with highest feature contribution weights sheds light on understanding local brain structural basis underlying emergence of distinctive functions manifested by daily behavior, enhancing our knowledge of brain-behavior relationships.

Motor scores from Bayley-III were not predicted reliably in this study possibly due to low variability and low signal-to-noise ratio (SNR) of cortical FA measurement in primary sensorimotor cortical regions associated with motor function. Cortical FA measurements at primary sensorimotor cortex are relatively low compared to those at other cortices (*Ball et al., 2013*; *Ouyang et al., 2019b*) and are barely above the noise floor, as primary sensorimotor cortex develops earlier compared to cortical regions associated with higher-order brain functions. Individual variability of cortical microstructure at primary sensorimotor cortex cannot be well captured with relatively low SNR for the cortical FA measurements at these regions. High individual variability enables reliable prediction. Low functional variability at primary sensorimotor cortex was found in a largely overlapped cohort in a separate study (*Xu et al., 2019*) from our group, and was reproducibly found in another cohort (http://developingconnectome.org). With strict exclusion criteria of participating cohort around birth, higher average and lower variance of motor scores than those of cognition or language scores play an important role in poor prediction of motor scores. Larger group variability in motor scores and larger sample size can offset the limitations elaborated above and enhance the prediction of motor scores.

Technical considerations, limitations, and future directions are discussed below. The capacity of cortical microstructural profile at birth to predict later behavior is substantial (*Figure 2*). We trained models to classify the low and normal outcomes. The ROC curves and accuracy measurements in *Figure 2—figure supplement 2* demonstrated high accuracy of the classification of low- and normal-outcome groups. Robustness of the prediction model was further tested against various factors. These factors included different cortical parcellation schemes (random and finer parcellations versus parcellation based on an atlas) for measuring feature vectors and individual age adjustment (*Figure 2—figure supplement 1*). Importantly, high performance of the prediction models is reproducible after taking above-mentioned factors into consideration. We used MAE as a metric to measure prediction errors as MAE is more robust to extreme or outlier values compared to other prediction errors such as mean square error or root mean squared error (*Willmott and Matsuura, 2005*). Despite relatively high dMRI resolution ($0.656 \times 0.656 \times 1.6$ mm$^3$) being used, partial volume effects (*Jeon et al., 2012*) cannot be ignored for measuring cortical FA. The partial volume effects are different across brain with thinner cortical regions more severely affected. To maximally alleviate the

partial volume effects and enhance the measurement accuracy, we adopted a 'cortical skeleton' approach (*Ouyang et al., 2019b*; *Yu et al., 2016*), demonstrated in the *Figure 1—figure supplement 4*, to measure cortical microstructure at the center or 'core' of the cortical plate. Although both preterm and term-born infants are included, none of them was clinically referred. All infants had been recruited solely for brain research and rigorously screened by a neonatologist and a pediatric neuroradiologist to exclude any infants with signs of brain injury (see Materials and methods for more details). To limit the effects of exposure to the extrauterine environment, this study was designed to make the interval between birth and scan age as short as possible. As a result, we did not find any significant correlation between birth age or MRI scan age and neurodevelopmental outcomes (see *Supplementary file 2*). The literature (*Bonifacio et al., 2010*) also indicated that the effects of premature birth on brain development are considered to be relatively trivial compared with the effects of brain injury and co-morbid condition which was not presented in any recruited infant due to strict exclusion criteria. However, including preterm infants is still considered a limitation as more pronounced neurodevelopmental deviation of children with preterm birth yet without any brain injury is possible after 2 years of age. Despite this limitation, this later deviation may not significantly affect conclusion of this study focused on 0–2 years of age. The established prediction here incorporating MRI of preterm and term born neonates scanned across 32–42 PMW could benefit future outcome prediction studies based on *in utero* MRI of the fetuses in the third trimester with recent advances *in utero* MRI techniques (e.g. *Khan et al., 2019*; *Thomason et al., 2013*; *Vasung et al., 2020*). With potential long-term effects on neurodevelopment in preterm neonates, *in utero* MRI may be a better choice for future studies on normal brain development in the third trimester than preterm neonates. Although we have taken many precautions to extract cortical FA measures and tested internal validity of our prediction analysis, several limitations will need to be addressed in future research. Despite the fact that relatively high performance of behavioral prediction at a group level was achieved with current cohort of infants, the prediction model will benefit from validation (e.g. *k*-fold cross-validation) and replication with an independent infant cohort of a larger sample size for generalization. Thus, prediction model with individual variability representing a general population from a much larger cohort is warranted in future research before this approach is effective for meaningfully predicting the outcome for a single individual. As indicated in *Figure 4b and c*, more baseline information resulted in improved prediction. Future research will also benefit from incorporating distinctive and complementary measurements from multimodal neuroimaging (e.g. *Kwon et al., 2014*; *Smyser et al., 2016*), including structural (i.e. T1- or T2-weighted), functional, and diffusion MRI. Genetic factors could also be incorporated. With these multimodal measurements, more advanced machine learning algorithms such as multi-kernel and deep learning need to be adopted and developed to further improve prediction. This study included a healthy cohort of infants for evaluating cortical microstructure for predicting future behavior. Such evaluation in the setting of pathology needs to be further validated. The observed neurodevelopmental outcomes were also contributed by unmeasured factors such as maternal age and tertiary educational level as well as other home environment variable following discharge from the hospital, all of which should be taken into consideration in the future prediction model.

In conclusion, whole-brain cortical FA at birth, encoding rich information of dendritic arborization and synaptic formation, could be reliably used for predicting neurodevelopmental outcomes of 2-year-old infants by leveraging individual variability of these measures. Feature contribution weight in cognitive or language prediction is heterogeneous across brain regions. The cortical regions contributing heavily to the prediction models exhibited distinguishable functional selectivity for cognition and language. Identifying regions with highest feature contribution weights offers preliminary findings on understanding local brain microstructural basis underlying emergence of future behavior, enhancing our knowledge of brain-behavior relationships. These findings also suggest that cortical microstructural information at birth may be potentially used for prediction of behavioral abnormality in infants with high risk for brain disorders early at a time when infants are pre-symptomatic in behavioral assessments and intervention may be most effective.

## Materials and methods

### Participants

The study was approved by the Institutional Review Board (IRB) at the University of Texas Southwestern Medical Center. A total of 107 neonates were recruited from the Parkland Hospital and scanned at Children's Medical Center at Dallas. Evaluable MRI was obtained from 87 neonates (58 M/29 F; post-menstrual ages at scan: 31.9–41.7 postmenstrual weeks (PMW); post-menstrual ages at birth: 26–41.4 PMW). All recruited infants were not clinically indicated. In other words, the infants in this study did not have medical reasons to be scanned with clinical MRI as they were considered healthy in routine medical care. They were recruited completely for research purpose which was studying the prenatal and perinatal human brain development. The benefit of MR scan could be that occasionally abnormality was found for some of these scanned infants after neuroradiologist's reading of their MRI. Data of these infants were then excluded from analysis in the current study. These neonates were selected through rigorous screening procedures by a board-certified neonatologist (LC) and an experienced pediatric radiologist, based on subjects' ultrasound, clinical MRI, and medical record of the subjects and mothers. Other exclusion criteria included evidence of bleeding or intracranial abnormality by serial sonography; the mother's excessive drug or alcohol abuse during pregnancy; periventricular leukomalacia; hypoxic–ischemic encephalopathy; Grade III–IV intraventricular hemorrhage; body or heart malformations; chromosomal abnormalities, lung disease, or bronchopulmonary dysplasia; necrotizing enterocolitis requiring intestinal resection or complex feeding/nutritional disorders; defects or anomalies of the brain; brain tissue dysplasia or hypoplasia; abnormal meninges; alterations in the pial or ventricular surface; or WM lesions. Informed parental consents were obtained from the subject's parent. More demographic information of the participants can be found in *Supplementary file 1*.

### Neonate brain MRI

All neonates were scanned with a 3T Philips Achieva System (ages at scan: 31.9–41.7 PMW). Neonates were fed before the MRI scan and wrapped with a vacuum immobilizer to minimize motion. During scan, all neonates were asleep naturally without sedation. Earplugs, earphones, and extra foam padding were applied to reduce the sound of the scanner. All 87 neonates underwent high-resolution dMRI and structural MRI scans. A single-shot echo-planar imaging (EPI) sequence with Sensitivity Encoding parallel imaging (SENSE factor = 2.5) was used for dMRI. Other dMRI imaging parameters were as follows: time of repetition (TR) = 6850 ms, echo time (TE) = 78 ms, in-plane field of view = $168 \times 168$ mm$^2$, in-plane imaging matrix = $112 \times 112$ reconstructed to $256 \times 256$ with zero filling, in-plane resolution = $0.656 \times 0.656$ mm$^2$ (nominal imaging resolution or acquisition resolution $1.5 \times 1.5$ mm$^2$), slice thickness = 1.6 mm without gap, slice number = 60, and 30 independent diffusion encoding directions with b value = 1000 s/mm$^2$. Two repetitions were conducted for dMRI acquisition to improve the SNR, resulting in a scan time of 11 min.

### Quality control and quality assurance of MRI

General MRI slice and slice-time integral measures for quality control (QC) were determined daily using ADNI and BIRN phantoms. Any systematic anomaly identified by significant deflections from normal variation was addressed immediately with technical support and/or the in-house MR physicist team. As is the laboratory practice, test-retest reliability of the MR imaging protocol was assessed with a four subject X four repeat estimation on intra- and inter-subject variation for quality assurance (QA).

### Measurement of cortical microstructure with brain MRI at birth

Diffusion tensor of each brain voxel was calculated with routine tensor fitting procedures. Diffusion MRI data sets from all neonates were preprocessed using *DTIstudio* (http://www.mristudio.org) (*Jiang et al., 2006*). To quantify head motion in each dMRI scan, all diffusion weighted image (DWI) volumes were aligned to the first stable image volume in the scan using automatic image registration (AIR) in *DTIStudio*. The volume-by-volume translation and rotation from the registration were calculated (*Ouyang et al., 2016*). As all MRI scans were conducted during neonates' natural sleep, in general the motion was very small during dMRI scans. With occasional abrupt movement during sleep,

DWI volumes with translation measurement larger than 5 mm or rotation measurement larger than 5° was determined as corrupted volumes. With 30 scanned DWI volumes and two repetitions, we accepted those scanned diffusion MRI data sets with less than 5 DWI volumes affected by motion. The second dMRI scan was immediately after the first dMRI scan. The affected volumes were replaced by the good volumes from another dMRI scan during postprocessing (*Huang et al., 2015*). After volume replacement, small motions in dMRI of all 46 infants who had a neurodevelopmental assessment at 2 years of age profiles were measured. Distribution of these motion measurements including translations and rotations are shown in *Figure 1—figure supplement 5*. The volume-by-volume translation range is 0–1.6 mm with average 0.78 mm and most of translations less than 1 mm. The volume-by-volume rotation range is 0–0.7° with an average of 0.18° and most of rotations less than 0.3°. Small motion and eddy current of dMRI for each neonate were corrected by registering all the DWIs to the non-diffusion weighted b0 image using a 12-parameter (affine) linear image registration with the AIR algorithm. Six elements of diffusion tensor were fitted in each voxel. Maps of FA derived from diffusion tensor were obtained for all neonates (*Figure 1*). DTI-derived FA maps were used to obtain the cortical skeleton FA measurements at specific cortical gyral region of interests (ROI) identified by certain gyral label from a neonate atlas (*Feng et al., 2019*). To alleviate partial volume effects, the cortical FA values were measured on the cortical skeleton, i.e. the center of the cortical mantle, demonstrated as green skeletons in the left panels in *Figure 1*. This procedure was elaborated in our previous studies (*Ouyang et al., 2019b*; *Yu et al., 2016*). The cortical skeleton was created from averaged FA maps in three age-specific templates at 33, 36, and 39 PMW due to dramatic anatomical changes of the neonate brain from 31.9 to 41.7 PMW. Based on the scan age, individual subject brain was categorized into three age groups at 33, 36, and 39 PMW, and registered to the corresponding templates using the registration protocol described in details in the literature (*Feng et al., 2019*; *Oishi et al., 2011*). By applying the skeletonization function in *TBSS* of *FSL* (http://fsl.fmrib.ox.ac.uk/fsl/fslwiki/TBSS), cortical skeleton of the 33 PMW or 36 PMW brain was extracted from the averaged cortical FA map and cortical skeleton of the 39 PMW brain was obtained with averaged cortical segmentation map due to low cortical FA in 39 PMW brains. The cortical skeleton in the 33, 36, and 39 PMW space was then inversely transferred to each subject's native space, to which the 52 cortical gyral labels of a neonate atlas (*Feng et al., 2019*) were also mapped to parcellate the cortex (*Figure 1*). *Figure 1—figure supplement 4* illustrates the workflow to parcellate and measure the cortical FA at the neonate cortical skeleton from a representative preterm and term born infant. Irregularly small yet significant offsets between the cortical skeleton and subject's cerebral cortex were widespread, due to imperfect inter-subject registration from transforming cortical skeletons to individual brains. Such offsets were addressed by using a fast-marching technique (details in *Jeon et al., 2012*; *Ouyang et al., 2019b*). With the fast-marching technique, FA at the voxels with highest gray matter tissue probability from segmentation of an individual subject, namely, 'core' cortical voxels, will be assigned to skeleton voxels sometimes deviating from 'core' (*Figure 1—figure supplement 4*-b5). As can be appreciated from *Figure 1—figure supplement 4* displaying a preterm brain at 33 PMW and a term born brain at 41 PMW with the same scale, there are almost no significant differences of partial volume effects on cortical FA measurements between preterm and term-born brains. By directly overlapping the cortical skeleton with the neonate atlas, the cortical skeleton was parcellated into 52 gyri. The FA measurement at each cortical gyrus was calculated by averaging the measurements on the cortical skeleton voxels with this cortical label. In this way, feature vectors consisting cortical FA values from 52 parcellated cortical gyri and measured at the cortical skeleton were obtained for the following SVR procedures.

## Neurodevelopmental assessments at 2 years of age

Out of 87 neonates with evaluable MRI scanned around birth, a follow-up neurodevelopmental assessment was obtained from 46 neonates (32 M/14 F, scan age of 36.7 ± 2.8 PMW) at their 2 years of age (20–29 months, 23.5 ± 2.3 months) corrected for prematurity, with gestational age taken into account. Cognitive, language, and motor development were assessed using *Bayley, 2006*. Specifically, the cognitive scale estimates general cognitive functioning on the basis of nonverbal activities (i.e. object relatedness, memory, problem solving, and manipulation); the language scale estimates receptive communication (i.e. verbal understanding and concept development) as well as expressive communication including the ability to communicate through words and gestures; and the motor scale estimates both fine motor (i.e. grasping, perceptual-motor integration, motor planning, and

speed) and gross motor (i.e. sitting, standing, locomotion, and balance) (*Bayley, 2006*). The Bayley-III is age standardized and widely used in both research and clinical settings. It has published norms with a mean (standard deviation) of 100 (15), with higher scores indicating better performance. This neurodevelopmental assessment was conducted by a certified neurodevelopmental psychologist, who was blinded to clinical details of infants as well as the neonate MR findings. Unlike cognitive, language, and motor scales reliably obtained using items administered to the child by a certified neurodevelopmental psychologist, other two scales from Bayley-III (social-emotional and adaptive scales) obtained from primary caregiver heterogeneous responses to questionnaires were not included in this study.

## Prediction of neurodevelopmental outcome with cortical FA as features

To determine whether cortical FA at birth could serve as a biomarker for individualized prediction of neurodevelopmental outcomes at 2 years of age, we performed pattern analysis using SVR algorithm implemented in *LIBSVM* (*Chang and Lin, 2011*). SVR is a supervised learning technique based on the concept of support vector machine (SVM) to predict continuous variables such as cognitive, language, or motor composite score from Bayley-III. LOOCV was adopted to evaluate the performance of the SVR model for each score. Cortical FA at birth from one individual subject was used as the testing data and the information of remaining 45 subjects including their cortical FA at birth and Bayley scores at 2 years of age were used as training data. In this procedure, the neurodevelopmental outcome of each infant was predicted from an independent training sample. Cortical FA measurements from 52 parcellated cortical gyri formed the feature vectors of each subject and were used as the SVR predictor. Feature vectors for all subjects were concatenated (*Feature vectors* in *Figure 1*) to obtain the input data for SVR prediction models with linear kernel function (*Figure 1*). Each feature represented by FA measurement at each cortical gyrus was independently normalized across training data. Only training data was used to compute the normalization scaling parameters, which were then applied to the testing data. After predicted continuous cognitive or language scores were estimated by the prediction model, Pearson correlation coefficient ($r$) and MAE between the actual and predicted continuous score were computed to evaluate cognition or language prediction models. The normalized feature contribution weights ($|w_i| / \sum |w_i|$ with $i$ indicating $i$th cortical gyrus) were calculated to represent contribution of all parcellated cortical gyri to the cognition or language prediction model. These normalized feature contribution weights of all parcellated cortical gyri in cognition or language prediction model were then mapped to the cortical surface to reveal heterogeneous regional contribution across entire cortex and distinguishable regional contribution distribution in a specific prediction model.

## Assessment of robustness of prediction

Permutation test was conducted to assess LOOCV prediction performance. Specifically, cognitive or language outcomes were randomly shuffled across subjects 1000 times. Prediction procedure was carried out with each set of randomized outcomes, generating null distributions. Pearson correlation was conducted for each set of randomized outcome. MAE between predicted and observed outcome from randomly shuffled distributions was also calculated. The p-values of observed correlation coefficient ($r$) value in LOOCV prediction, calculated as the ratio of number of permutation tests with correlation coefficient greater than observed $r$ value over number of all permutation tests, are the probability of observing the reported $r$ values by chance. Similarly, the p-values of MAE in LOOCV prediction, calculated as the ratio of number of permutation tests with MAE value lower than observed MAE value over number of all permutation tests, are the probability of observing the reported MAE by chance.

To investigate the effect of cortical parcellation schemes on the cortical FA measures in prediction model, various cortical parcellation schemes, including 52 cortical regions from the neonate atlas labeling (*Feng et al., 2019*), 128, 256, 512, and 1024 randomly parcellated cortical regions with equal size (*Zalesky et al., 2010*) were tested. For each parcellation scheme, averaged value of skeletonized FA measurements in each cortical ROI was used as a feature in the SVR model to test prediction performance. To address a possible confounding factor of various neonate gestational ages at scan, we evaluated if the prediction performance of cortical FA measures remained high after controlling for ages at scan following the age correction methods described in the literature

(*Dukart et al., 2011*). Specifically, age effect in the cortical FA of each gyrus (*Ouyang et al., 2019b*) or each parcellated ROI was adjusted with a biphasic piecewise linear regression between cortical FA and age with inflection point in the biphasic piecewise linear model at 36 PMW. The cortical FA residuals in the biphasic piecewise linear regression model, considered as age adjusted cortical FA measures, were then used as features in SVR models for predicting the Bayley-III scores. To validate the capability of cortical FA in behavioral predictions, individual's Bayley composite scores were categorized into normal (>85) and low scores (≤85). Cortical FA measures were used as features to classify each subject's score into normal- or low-score groups using SVM algorithm with LOOCV. Classification accuracy and area under the receiver operating characteristic (ROC) curves were used to evaluate the performance of classification models.

## Bootstrap analysis for assessing reproducibility of top 10 cortical regions identified by LOOCV analysis

We used a bootstrap sampling approach to assess reproducibility of top 10 cortical regions where microstructural measures contributed most to predicting each outcome in LOOCV analysis. Specifically, we randomly selected 90% of the total 46 samples 1000 times. We then built cognition or language prediction model with each set of selected samples and identified top 10 cortical regions with highest contribution to the prediction of cognitive or language outcome. In each of 1000 bootstrap resamples, if any cortical region was identified as top 10 cortical regions contributing to predicting cognition outcome, the count for this specific cortical region was added by 1. After testing with 1000 resamples, the percentage of a certain cortical region was calculated as the total count for this region divided by 1000. In this way, a percentage map of all cortical gyri for predicting cognitive outcome can be created. The same procedure was repeated with 1000 bootstrap resamples for predicting language outcome. If the top 10 cortical regions where microstructural measures contributed most to predicting each outcome in LOOCV analysis (*Figure 3*) overlaps with the cortical regions with high percentage, it indicated that the top 10 cortical regions identified by LOOCV analysis were highly reproducible.

## Permutation tests to assess distinguishable regional contribution to predicting cognitive or language outcomes

To quantify the extent of distinction between the set of top 10 cortical regions in cognition prediction model and the set of top 10 cortical regions in language prediction model, we defined a nonoverlapping index as the number of nonoverlapped regions between these two sets divided by the number of regions in their union set. This nonoverlapping index ranges from 0 to 1, with one indicating completely distinctive sets of regions and 0 indicating completely same sets of regions. A permutation test was used to evaluate the statistical significance of the observed nonoverlapping index. The null hypothesis is that the observed nonoverlapping index from predicting two different outcomes is not different from a distribution of nonoverlapping index calculated from predicting same (cognitive or language) outcome. The null distribution of nonoverlapping indices was generated by calculating 2070 nonoverlapping indices with each corresponding to one of 1035 pairs of cognitive-cognitive outcome or one of 1035 pairs of language-language outcome using leave-one-out resamples. The p-value of reported nonoverlapping index is the probability of observing the reported nonoverlapping index by chance and was calculated as the number of permutations with higher index value than reported index divided by the number of total permutations. We also conducted a more strict permutation test by increasing variability of the resamples. Specifically, the bootstrap resamples used in the section of 'bootstrap analysis for assessing reproducibility of top 10 cortical regions identified by LOOCV analysis' above was adopted to generate another null distribution of nonoverlapping indices and calculate the p-value of observed nonoverlapping index by using the same procedures described above.

## Prediction of neurodevelopmental outcome with WM FA only and combined cortical and WM FA as features

WM FA only and combined cortical and WM FA were tested for predicting neurodevelopmental outcomes. WM skeleton FA values at the core were measured to alleviate the partial volume effects (left panel in *Figure 4a*). WM skeleton was further parcellated into 40 tracts with the tract labeling

transformed from a neonate atlas (*Feng et al., 2019*). Details of tract-wise FA measurement at the WM skeleton were described in our previous publication (*Huang et al., 2012*). WM FA measurements of the 40 tracts were used to generate the feature vectors of each subject. In addition, cortical FA measurements of the 52 cortical regions and WM FA measurements of the 40 tracts were combined to generate the feature vector of each subject. Similar to the procedures of predicting neurodevelopmental outcomes with cortical FA feature vectors, these WM FA feature vectors only or combined cortical and WM FA feature vectors were the input of the SVR predictor with LOOCV for predicting neurodevelopmental outcome. To evaluate the generalizability of the prediction model, a threefold cross validation analysis was also applied. Unlike LOOCV trained on the data from all but one participant, threefold cross validation left out 33% of the participant (15 of 46) data for testing and was trained on remaining 67% of the data.

## Acknowledgements

This study was sponsored by NIH R01MH092535, R01MH092535-S1, and U54HD086984. We would like to thank Brittany C Bennett at the Children's Hospital of Philadelphia for her contribution to the schematic depiction.

## Additional information

### Funding

| Funder | Grant reference number | Author |
|--------|------------------------|--------|
| National Institutes of Health | MH092535 | Hao Huang |
| National Institutes of Health | MH092535-S1 | Hao Huang |
| National Institutes of Health | HD086984 | Hao Huang |

The funders had no role in study design, data collection and interpretation, or the decision to submit the work for publication.

### Author contributions

Minhui Ouyang, Conceptualization, Resources, Data curation, Software, Formal analysis, Validation, Investigation, Visualization, Methodology, Writing - original draft, Writing - review and editing; Qinmu Peng, Software, Validation, Methodology, Writing - review and editing; Tina Jeon, Data curation, Formal analysis, Investigation, Project administration, Writing - review and editing; Roy Heyne, Lina Chalak, Resources, Data curation, Project administration, Writing - review and editing; Hao Huang, Conceptualization, Resources, Data curation, Formal analysis, Supervision, Funding acquisition, Validation, Investigation, Visualization, Methodology, Writing - original draft, Project administration, Writing - review and editing

### Author ORCIDs

Hao Huang https://orcid.org/0000-0002-9103-4382

### Ethics

Human subjects: Informed parental consents were obtained from the subject's parent. The Institutional Review Board of both University of Texas Southwestern Medical Center (CR00009778 / STU012012-132) and Children's Hospital of Philadelphia (IRB 15-011775) approved study procedures.

### Decision letter and Author response

Decision letter https://doi.org/10.7554/eLife.58116.sa1
Author response https://doi.org/10.7554/eLife.58116.sa2

# Additional files

## Supplementary files

• Supplementary file 1. Demographics of participating subjects who went through MRI scans at birth and was assessed with Bayley tests at their 2 years of age. [a]C for C-section and V for vaginal birth; [b]B for breast-feeding and F for formula; wks: postmenstrual weeks.

• Supplementary file 2. Mean and standard deviation of the composite scores from the 2 years. Bayley-III neurodevelopmental assessments of the 46 infants as well as *r* and p-values of the correlation between one of the composite scores and a specific age (birth age, scan age, or Bayley-III assessment age) in both preterm and term born infant groups.

• Supplementary file 3. Normalized feature contribution weights of cortical fractional anisotropy measures from each gyrus in cognitive and language prediction models. The top 10 highest weights in each model are marked in bold.

• Transparent reporting form

## Data availability

Neonate MRI datasets are publicly available and can be freely downloaded from brainmrimap.org (a public website maintained by Huang lab). Behavioral datasets are available in the supplemental information of this publication. Source codes used for prediction are available from first author's github repository (https://github.com/MHouyang/Prediction-of-neurodevelopmental-outcome ; copy archived at https://archive.softwareheritage.org/swh:1:rev:e3506bfbfa03db27651-b1803e5cb662b623f5360/).

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
