## [Decision Letter]

**Acceptance summary:**

This paper uses brain imaging in neonates to show that brain structure at birth can predict behaviour up to 2 years later. This work provides an important link between brain health at birth and later cognitive development.

**Decision letter after peer review:**

Thank you for submitting your article "Diffusion-MRI-based regional cortical microstructure at birth for predicting neurodevelopmental outcomes of 2-year-olds" for consideration by *eLife*. Your article has been reviewed by three peer reviewers, one of whom is a member of our Board of Reviewing Editors, and the evaluation has been overseen by Richard Ivry as the Senior Editor. The reviewers have opted to remain anonymous.

The reviewers have discussed the reviews with one another and the Reviewing Editor has drafted this decision to help you prepare a revised submission.

Summary

This manuscript focuses on trying to predict cognitive assessment scores, collected at 18 months of age, from neonatal cortical structure. In 46 neonates, they use support vector regression with leave-one-out cross-validation to model the multivariate relationship between cortical microstructure and later scores. The authors are able to calculate predicted individual cognitive and language scores that correlate with actual scores.

All reviewers appreciate the challenge in collecting these kinds of data and the cross-validation procedure used for outcome prediction. However, we shared a series of concerns that must be addressed before we can consider publication.

Essential revisions

– We are concerned about the representativeness and size of the sample, and therefore how generalisable these results might be to the larger neonatal population. The infants are both born and scanned across a very wide age range, 32-42 postnatal weeks. This means that some of the infants were both very preterm at birth and at scan, with very different cortical microstructure (seen in the supplementary figures). There is also very different partial voluming in these two ages simply due to brain volume differences. Moreover, the study population is not described very well. We cannot discern what the relative proportions of preterm and term infants are. The authors only report that the gestational age ranged from 26 to 41 weeks and that images were obtained at postmenstrual ages ranging from 31 to 41 weeks. More detail is needed regarding the study population and who was imaged when. In addition, more than half of the recruited infants didn't finish the study, and it is important to know if there were differences between those subjects who were apparently lost to follow up and those that weren't.

– The mixing of term and preterm groups in the analysis raises an interpretation issue. These two populations aren't equivalent, and few would argue that preterm infants are "normal" despite having normal conventional neuroimaging studies. It is conceivable that the findings of the study are driven by abnormalities in the preterm infants, and are thereby an indication of areas which are most commonly injured in preterm infants. We suggest the authors consider i) comparing findings of preterm with term infants and ii) evaluating outcome correlations for these populations separately. An alternative would be to examine term-equivalent scans for all participants.

– Please clarify the extent to which the correlations between real and predicted scores driven by extreme scorers.

– The authors have demonstrated the quite dramatic changes that occur over this period (Ouyang et al., 2019). It's difficult to see how you could combine datasets that have the transient anisotropic features in cortex (early) and more isotropic mature features (late) without testing on a hold out. Linear age adjustment doesn't really help in this context as the changes are nonlinear. As an example, the positive and negative scores at both ends of the tails of the Bailey-III cognitive scores are preterm and term born infants respectively which again makes me worry about the confound of age in the results of the regression.

– The average assessment score is nearly a full standard deviation below what you'd expect in a typical population at the same age, averaging 85-90 in the three subscales. Is there a reason the average is so low?

– The predicted scores are also in a very restricted range – although the raw scores range from 65-110 (cognitive) and 55-110 (language) the predictions seem to only range from 80-90. As you move away from the mean, the absolute error increases substantially.

– More generally, the analysis provides a useful proof-of-principle that early FA measures can predict later outcomes, but the predictions are not sufficiently accurate for clinical use. The MAE for the models is around 1 SD of the cognitive scores, and the accuracy using performance cut-offs was 61%/76%. Although sensitivity/specificity are not reported, FA does not seem to be a highly sensitive marker of these outcomes (as suggested in text). This approach may work at a population level but is not particularly effective for meaningfully predicting the outcome for a single individual. A more measured tone in describing the findings and their limitations would provide a more balanced account of the findings.

– With samples sizes this small there is substantial risk of overestimation of the accuracy of any machine learning techniques. The sample size is small, but 5-fold or 10-fold CV is possible here – see Poldrack et al. 10.1001/jamapsychiatry.2019.3671

– White matter FA showed a similar prediction accuracy to cortical FA. This raises questions about specificity. Could simpler measures (e.g., T1 or T2 signal) provide comparable prediction? If so, this may challenge the biological interpretation that the prediction derives from FA's capacity to measure cortical microstructure. Perhaps the authors could further examine this issue by comparing the relative prediction efficacy of simpler and FA-derived measures and looking at how strongly cortical and white matter measures correlate with each other. Is there anything to be gained by combining them?

– Please clarify how motion corruption of the DWI data was assessed and determined?

– Please explain reasons for participant dropout over the 2 year follow-up.

[Editors' note: further revisions were suggested prior to acceptance, as described below.]

Thank you for resubmitting your work entitled "Diffusion-MRI-based regional cortical microstructure at birth for predicting neurodevelopmental outcomes of 2-year-olds" for further consideration by *eLife*. Your revised article has been evaluated by Richard Ivry (Senior Editor) and a Reviewing Editor.

The manuscript has been improved but there are some remaining issues that need to be addressed before acceptance, as outlined below. We emphasise that these issues must be addressed to the satisfaction of the reviewers before the manuscript can be accepted.

The first major concern is that the motion threshold is very large (>3 times voxel dimensions) and the replacement scan seems to be taken from a different session. The eddy correction approach is also out of date and seems to just be an affine registration (no outlier rejection like in tortoise, eddy, shard). The authors need to comprehensively demonstrate that the results are not driven by motion-related artefact.

Second, it may not be appropriate to classify prematurely-born infants as "normal." Although the MRI may appear normal at early neurodevelopmental follow up, many cognitive issues aren't detectable until these children reach school age. This should be identified as a weakness in the Discussion.

[Editors' note: further revisions were suggested prior to acceptance, as described below.]

Thank you for sending your article entitled "Diffusion-MRI-based regional cortical microstructure at birth for predicting neurodevelopmental outcomes of 2-year-olds" for peer review at *eLife*. Your article is being evaluated by three peer reviewers, and the evaluation is being overseen by a Reviewing Editor and Richard Ivry as the Senior Editor.

The reviewers feel that the issue of head motion has not been fully addressed. The reviewers requested that you comprehensively show that motion cannot explain the findings. Showing the distribution of motion estimates in the sample is not sufficient. For publication, we would require a stronger demonstration that motion does not contaminate or confound the predictions by, for example, examining correlations between motion estimates (e.g., framewise displacement) and the outcome/connectivity measures used in the analysis.

---

## [Author Response]

Essential revisions– We are concerned about the representativeness and size of the sample, and therefore how generalisable these results might be to the larger neonatal population. The infants are both born and scanned across a very wide age range, 32-42 postnatal weeks. This means that some of the infants were both very preterm at birth and at scan, with very different cortical microstructure (seen in the supplementary figures). There is also very different partial voluming in these two ages simply due to brain volume differences. Moreover, the study population is not described very well. We cannot discern what the relative proportions of preterm and term infants are. The authors only report that the gestational age ranged from 26 to 41 weeks and that images were obtained at postmenstrual ages ranging from 31 to 41 weeks. More detail is needed regarding the study population and who was imaged when. In addition, more than half of the recruited infants didn't finish the study, and it is important to know if there were differences between those subjects who were apparently lost to follow up and those that weren't.

We thank the reviewer for this comment. We acknowledge that a study with larger sample size may further improve generalization of the proposed model. On the other hand, as elaborated in the response to general comment, we would think a very wide age range will benefit generalization given that these neonates are free from perinatal brain injuries. The model can be applied to future *in utero* MRI where wide age range could be more likely. In the present study, the statistical analyses demonstrated no significant difference in any of the outcome scores, namely cognitive, language and motor scores, between the preterm and term infants (all *p*>0.3 in the panel (b) of updated Figure 1—figure supplement 3).

We are very aware of the partial volume effects in measuring cortical microstructure and have developed robust method to measure the cortical fractional anisotropy (FA) at the core of the cortical mantle, as demonstrated in added Figure 1—figure supplement 4. This method has been successfully and consistently implemented in our previous studies (e.g. Ouyang et al., 2019; Yu et al., 2016) too. Details of measuring cortical FA at the core (or skeleton) of the cortical ribbon can be found in the subsection “Measurement of cortical microstructure with brain MRI at birth” in the Materials and methods section. Briefly, skeleton voxels or “core” of the cerebral cortical mantle were obtained. Irregularly small yet significant offsets between the cortical skeleton and subject’s cerebral cortex were widespread, due to imperfect inter-subject registration from transforming cortical skeletons to individual brains. Such offsets were addressed by using a fast marching technique (details in our publication, Jeon et al., 2012; Ouyang et al., 2019). With fast marching, FA at the voxels with highest gray matter tissue probability from segmentation of an individual subject, namely, “core” cortical voxels, will be assigned to skeleton voxels sometimes deviating from “core” (Figure 1—figure supplement 4-a5, b5). As can be appreciated from Figure 1—figure supplement 4 using same scale to display a preterm brain at 33 postmenstrual weeks (PMW) and a term born brain at 41PMW, there are almost no significant differences of partial volume effects on cortical FA measurements between preterm and term-born brains.

Supplementary file 1 has been updated to describe subdivided preterm and term born populations. Supplementary file 2 has also been updated to demonstrate no significant correlation between the neurodevelopmental outcome and age (birth age or scan age) in subdivided preterm or term born populations. We intended to follow up all subjects undergoing MRI around birth and conduct neurodevelopmental assessment at their 2 years of age. Study staff worked diligently for that. They made biannual phone calls and sent birthday cards to stay in contact with families who showed interests to be contacted about follow-up study to ensure contact information remained current. All subjects who did not participate 2-year follow-up were those lost to follow up. Relatively moderate follow-up rate might be related that neither recruited preterm nor term born neonates had any perinatal brain injury or were clinically indicated. Motivation of participating 2-year follow-up for these parents may not be as high as parents of infants scanned due to clinical indication in other literature.

– The mixing of term and preterm groups in the analysis raises an interpretation issue. These two populations aren't equivalent, and few would argue that preterm infants are "normal" despite having normal conventional neuroimaging studies. It is conceivable that the findings of the study are driven by abnormalities in the preterm infants, and are thereby an indication of areas which are most commonly injured in preterm infants. We suggest the authors consider i) comparing findings of preterm with term infants and ii) evaluating outcome correlations for these populations separately. An alternative would be to examine term-equivalent scans for all participants.

Please see responses to the general comments. No injury was found in any preterm infants. None of the preterm infants were recruited due to clinical indication like most of relevant literature. Except preterm, the preterm infants were considered healthy and recruited solely for research purpose. All participating neonates were carefully screened by a neonatologist and a pediatric neuroradiologist to ensure “normality”. This study was also designed to make the interval between the birth and scan time as short as possible to limit the effects of exposure to the extrauterine environment. Furthermore, following reviewers’ suggestion, we compared the neurodevelopmental outcomes between preterm and term infants and found no statistically significant difference between these two populations (all *p*>0.3), shown in Figure 1—figure supplement 3B. We also evaluated outcome correlations with different age indices (i.e. birth age, MRI scan age and Bayley-III exam age) for preterm and term born infants separately and found no significant correlations (all *p*>0.1) in updated Supplementary file 2. Considering all factors above, it does not seem necessary to divide the neonates into preterm and term born populations for prediction analysis. Limited total sample size also prevented us from further subdividing the sample into two groups for prediction analysis. As we do not have term-equivalent scans for preterm neonates (exactly for limiting the effects of exposure to the extrauterine environment), we cannot exam term-equivalent scans for all participants.

– Please clarify the extent to which the correlations between real and predicted scores driven by extreme scorers.

We thank the reviewer for this comment. The correlations between real and predicted scores should not be driven by extreme scorers. In the present study we used mean absolute error (MAE) as a metric to measure prediction errors of our method. MAE is more robust to extreme/outlier values in the data compared to other prediction errors such as mean square error and root mean squared error (Willmott and Matsuura, 2005). This clarification has been added to the second to the last paragraph in the Discussion section.

To further confirm our results were not significantly affected by extreme scorers, we removed the smallest score values in the cognitive and language and re-ran correlation analysis between real and predicted scores. We found that all correlations remain significant (MAE = 5.74, *r*=0.496, *p* = 5x10^-4^ for cognitive and MAE = 6.95, *r* = 0.461, *p* = 1.5x10^-3^ for language).

– The authors have demonstrated the quite dramatic changes that occur over this period (Ouyang et al., 2019). It's difficult to see how you could combine datasets that have the transient anisotropic features in cortex (early) and more isotropic mature features (late) without testing on a hold out. Linear age adjustment doesn't really help in this context as the changes are nonlinear. As an example, the positive and negative scores at both ends of the tails of the Bailey-III cognitive scores are preterm and term born infants respectively which again makes me worry about the confound of age in the results of the regression.

We thank the reviewer for this comment. We agree that changes of cortical FA are not linear across the entire studied period, but rather biphasic piecewise linear with an inflection point at 36PMW (Ouyang et al., 2019). To better correct the age effect in cortical FA features in this revised manuscript, we adopted the same method to quantify cortical FA time courses (more details in Ouyang et al., 2019). The biphasic piecewise linear model below was used to fit the cortical FA of each gyrus or each parcellated region of interest (ROI) and age during 31-36PMW and 36-42PMW:

Equation 1 31-36 week FAi,j=β1,i+β2,itj+δi,j

Equation 2 36-42 week FAi,j=β3,i+β4,itj+γi,j

Where *t_j_*is the post-menstrual age of the jth subject; *β_1,i_*, *β_3,i_* are the intercepts and *β_2,i_*, and *β_4,I_*are the slopes for FA of the ith cortical ROI; δ*_i,j_* and γ*_i,j_* are the error term for cortical FA measurement from 31-36PMW and from 36-42PMW, respectively. We corrected the age effect of cortical FA features of each gyrus or each parcellated ROI, FAi,jcorrect, based on above biphasic piecewise linear model, following the age correction methods described in the literature (Dukart et al., 2011):

31-36 week FAi,jcorrect=FAi,j+β2,i*(36−tj)

36-42 week FAi,jcorrect=FAi,j+β4,i*(36−tj)

After above-mentioned adjustment for the age effect, correlation between the predicted and actual cognitive or language scores is still significant (*p<0.05*) with original parcellation of 52 cortical regions and across other tested cortical parcellation schemes (128, 256, 512 and 1024 cortical parcels). Prediction performances before and after adjustment for the age effect are demonstrated in the updated Figure 2—figure supplement 1B-1C.

– The average assessment score is nearly a full standard deviation below what you'd expect in a typical population at the same age, averaging 85-90 in the three subscales. Is there a reason the average is so low?

The slightly lower average score may be related that samples are not diversified or large enough. All subjects are from a single county’s public hospital and recruitment was within 3 years. We would also make it clear that in this study average composite scores 85.7 – 91.2 in the three subscales are completely within 1 standard deviation from Bayley-III norms (mean = 100; standard deviation = 15). These subjects are still considered normal (Bayley, 2006).

– The predicted scores are also in a very restricted range – although the raw scores range from 65-110 (cognitive) and 55-110 (language) the predictions seem to only range from 80-90. As you move away from the mean, the absolute error increases substantially.

Restricted range of predicted score probably is mainly driven by restricted range of actual score. Although the actual score range is relatively wide, most of actual scores are clustered in 80 to 100 (Figure 1—figure supplement 3). Restricted predicted score range is also related to cortical parcellation scheme (Figure 2—figure supplement 1A). Especially using parcellation scheme with 128 random cortical parcels yields wide range of predicted scores. In addition, combining regional cortical FA and white matter FA measures in prediction (suggested by Essential revisions #9 below) also yields a wider range in predicted value. All these results suggest that larger individual variability of neurodevelopmental scores from a larger sample and more comprehensive information from multimodal neuroimaging (also see response to Essential revisions #9) will improve prediction performance and reduce prediction errors. Such discussion has already been included in the Discussion section.

– More generally, the analysis provides a useful proof-of-principle that early FA measures can predict later outcomes, but the predictions are not sufficiently accurate for clinical use. The MAE for the models is around 1 SD of the cognitive scores, and the accuracy using performance cut-offs was 61%/76%. Although sensitivity/specificity are not reported, FA does not seem to be a highly sensitive marker of these outcomes (as suggested in text). This approach may work at a population level but is not particularly effective for meaningfully predicting the outcome for a single individual. A more measured tone in describing the findings and their limitations would provide a more balanced account of the findings.

We agree with the reviewer that before the predictions can be sufficiently accurate for clinical use, testing with a larger sample is needed. Sensitivity/specificity has been provided in Figure 2—figure supplement 2. We also agree that the approach shows effectiveness of prediction at the level of certain subgroups, but is not effective enough for accurate prediction of the outcome at the level of a single individual. As suggested by the reviewer, to provide a more balanced account of the finding, we have toned down the discussion significantly in the first and second to the last paragraph in the Discussion section.

– With samples sizes this small there is substantial risk of overestimation of the accuracy of any machine learning techniques. The sample size is small, but 5-fold or 10-fold CV is possible here – see Poldrack et al. 10.1001/jamapsychiatry.2019.3671

We thank the reviewer for this comment. Although *k*-fold cross-validation (CV) is ideal and can be applied to this prediction model, leave-one-out cross validation approach has also been widely used in studies reviewed in Poldrack et al., 2020. From Figure 3A in Polrack et al., 2020, number of leave-one-out studies was similar to number of k-fold studies.

As elaborated in response to Essential revision #6, #7 and #9, this manuscript emphasizes the capability of cortical microstructure measures of predicting future neurodevelopmental outcomes, not perfect prediction itself, as cortical microstructure measures have been rarely used for prediction. Following reviewer’s suggestion, we conducted a 3-fold cross validation by using cortical FA only, white matter (WM) FA only, and combined cortical and WM FA. As shown in Author response table 1, with combined cortical and WM FA, the predicted cognitive and language scores remained significantly correlated with the actual scores (all *p*<0.05) for all 3 folds, consistent to better prediction of combined cortical and WM FA in revised Figure 4B and 4C. Although using cortical FA or WM FA only did not pass the 3-fold CV, cortical FA features performed slightly better than WM FA features in terms of correlation coefficients. We have added these findings in the last paragraph of the Results section and the second paragraph of the Discussion section.

**Author response table 1. resptable1:** Mean correlation coefficient and the corresponding p values between predicted and actual scores in the 3-fold cross validation. * in the table indicates significant (p<0.05) correlation.

Features in prediction	Cognitive	Language		
	*r*	*p*	*r*	*p*
Cortical FA	0.508	0.053	0.479	0.071
White matter (WM) FA	0.491	0.063	0.459	0.085
Combined cortical and WM FA	0.635	**0.01 ***	0.592	**0.02 ***

– White matter FA showed a similar prediction accuracy to cortical FA. This raises questions about specificity. Could simpler measures (e.g., T1 or T2 signal) provide comparable prediction? If so, this may challenge the biological interpretation that the prediction derives from FA's capacity to measure cortical microstructure. Perhaps the authors could further examine this issue by comparing the relative prediction efficacy of simpler and FA-derived measures and looking at how strongly cortical and white matter measures correlate with each other. Is there anything to be gained by combining them?

We thank the reviewer for raising this question. It is our understanding that any measurement reflecting individual brain maturational level including T1 or T2 signal, white matter microstructure, could possibly be used for predicting neurodevelopmental outcome. The main purpose of this manuscript is not to claim that cortical FA is the “only” index for predicting neurodevelopmental outcome. Instead, we consider revealing capability of cortical FA of predicting neurodevelopmental outcome (probably for the first time) the main and novel contribution of this manuscript. These different brain measures reflect distinct brain developmental properties. For example, simpler measures from structural T1-weighted (T1w) or T2-weighted (T2w) images usually indicate macrostructural measurements such as cortical volume or thickness. White matter FA measures indicate white matter (not directly cerebral cortical) microstructure, although they might be related to cortical FA measures.

Following reviewer’s comment, we combined cortical and white matter FA measures to predict outcomes with results shown in Figure 4B and 4C. Higher performances were achieved in predicting outcomes by using combined features, demonstrated by higher correlation coefficient values between the predicted and actual scores in Figure 4B and 4C. We also added discussion about combining features to improve prediction performance in the Results and Discussion sections.

– Please clarify how motion corruption of the DWI data was assessed and determined?

To quantify head motion in each dMRI scan, all diffusion weighted image (DWI) volumes were aligned to the first stable image volume in the scan using automatic image registration (AIR) in DTIStudio (Jiang et al., 2006). The volume-by-volume translation and rotation from the registration were calculated. DWI volumes with translation measurement larger than 5 mm or rotation measurement larger than 5 degree was determined as corrupted volumes. With 30 scanned diffusion weighted image (DWI) volumes and 2 repetitions, we accepted those scanned diffusion MRI datasets with less than 5 DWI volumes affected by motion. The affected volumes were replaced by the good volumes of another DTI repetition during postprocessing. We have added these details in the Materials and methods section.

– Please explain reasons for participant dropout over the 2 year follow-up.

As elaborated in the response to general comment and Essential revision 2, we intended to follow up all subjects undergoing MRI around birth and conduct neurodevelopmental assessment at their 2 years of age. All subjects who did not participate 2-year follow-up were those lost to follow up. Relatively moderate follow-up rate might be related that neither recruited preterm nor term born neonates had any perinatal brain injury or were clinically indicated. Motivation of participating 2-year follow-up for these parents may not be as high as parents of infants scanned due to clinical indication in other literature.

[Editors' note: further revisions were suggested prior to acceptance, as described below.]

The first major concern is that the motion threshold is very large (>3 times voxel dimensions) and the replacement scan seems to be taken from a different session. The eddy correction approach is also out of date and seems to just be an affine registration (no outlier rejection like in tortoise, eddy, shard). The authors need to comprehensively demonstrate that the results are not driven by motion-related artefact.

We thank the reviewer for this comment. The second diffusion MRI (dMRI) scan is immediately after the first dMRI scan. The affected dMRI volumes were replaced by the good volumes from another dMRI scan in the same session. As all dMRI scans were conducted during neonates’ natural sleep, in general the motion was *very* small during scans. Motion threshold of 5 mm or 5 degrees was used only for discarding diffusion volumes with large motions due to occasional abrupt movements during sleep. The distribution of motion measurements including translation and rotation of dMRI scans in this cohort of 46 neonates who had a neurodevelopmental assessment at 2 years of age was demonstrated in Figure 1—figure supplement 5. The volume-by-volume translation range is 0 to 1.6mm with average 0.78mm and most of translations less than 1mm. The volume-by-volume rotation range is 0 to 0.7 degrees with average of 0.18 degrees and most of rotations less than 0.3 degrees. The information above was added to Materials and methods section.

In Author response image 1, axial b0 image from raw dMRI of a representative neonate brain in this study and that of a 3-year-old child brain scanned with the same imaging protocol are shown. Yellow arrows highlight relatively mild distortion of b0 image (left) of the neonate brain and severe distortion of b0 image (right) of the 3-year-old brain. Both b0 images are before eddy current correction and at the level of eyeball. To correct for the small head motion and eddy current distortions, we only used the 12-parameter (affine) registration by registering all the diffusion weighted images to the b0 image. Although we did not use automatic outlier rejection methods (e.g. Tortoise, eddy mentioned by the reviewer), we manually inspected all image slices and volumes and found no outliers due to severe eddy current distortions. Meanwhile, since no dMRI with opposite phase encoding directions was acquired, we could not apply advanced eddy current distortion correction technique requiring dMRI of both AP (anterior-posterior) and PA (posterior-anterior) directions.

**Author response image 1. sa2fig1:** The axial b0 image of a representative neonate (left) and 3-year-old (right) brain scanned with same dMRI imaging protocol.

Second, it may not be appropriate to classify prematurely-born infants as "normal." Although the MRI may appear normal at early neurodevelopmental follow up, many cognitive issues aren't detectable until these children reach school age. This should be identified as a weakness in the Discussion.

Following reviewer’s suggestion, we have removed “normal” in the first paragraph in the Results and Materials and methods section and added discussion to identify inclusion of prematurely-born infants as a limitation in the Discussion section.

[Editors' note: further revisions were suggested prior to acceptance, as described below.]

The reviewers feel that the issue of head motion has not been fully addressed. The reviewers requested that you comprehensively show that motion cannot explain the findings. Showing the distribution of motion estimates in the sample is not sufficient. For publication, we would require a stronger demonstration that motion does not contaminate or confound the predictions by, for example, examining correlations between motion estimates (e.g., framewise displacement) and the outcome/connectivity measures used in the analysis.

To comprehensively demonstrate our prediction results were not driven by motion-related artefact, we examined the correlation between motion (quantified by mean framewise displacement suggested in the Editor’s letter) and regional fractional anisotropy (FA) values from cerebral cortex and white matter (Figure 4—figure supplement 1a). After false discovery rate (FDR) correction, no significant correlation between motion estimates and FA values of any of 92 brain regions was found. We also investigated motion effects on prediction results and found that high performance of prediction models was not affected after statistically controlling for motion estimates in cortical and white matter FA measures in the predication models, as shown in Figure 4—figure supplement 1b. Figure 4—figure supplement 1A and 1B collectively demonstrated that motion did not contaminate or confound the significant predication found in this study. The response above has been added at the end of Results section.